# Game-theoretic agent-based modelling of micro-level conflict: Evidence from the ISIS-Kurdish war

**Olivia Macmillan-Scott**[1]*, **Akin Ünver**[2], **Mirco Musolesi**[1,3,4]

**1** Department of Computer Science, University College London, London, United Kingdom, **2** Department of International Relations, Ozyegin University, Istanbul, Turkey, **3** The Alan Turing Institute, London, United Kingdom, **4** Department of Computer Science and Engineering, University of Bologna, Bologna, Italy

\* olivia.macmillan-scott.16@ucl.ac.uk

## Abstract

This article delves into the dynamics of a dyadic political violence case study in Rojava, Northern Syria, focusing on the conflict between Kurdish rebels and ISIS from January 1, 2017, to December 31, 2019. We employ agent-based modelling and a formalisation of the conflict as an Iterated Prisoner's Dilemma game. The study provides a nuanced understanding of conflict dynamics in a highly volatile region, focusing on microdynamics of an intense dyadic strategic interaction between two near-equally- powered actors. The choice of using a model based on the Iterated Prisoner's Dilemma, though a classical approach, offers substantial insights due to its ability to model dyadic, equally-matched strategic interactions in conflict scenarios effectively. The investigation primarily reveals that shifts in territorial control are more critical than geographical or temporal factors in determining the conflict's course. Further, the study observes that the conflict is characterised by periods of predominantly one-sided violence. This pattern underscores that the distribution of attacks, and target choices are a more telling indicator of the conflict nature than specific behavioural patterns of the actors involved. Such a conclusion aligns with the strategic implications of the underlying model, which emphasises the outcome of interactions based on differing aggression levels. This research not only sheds light on the conflict in Rojava but also reaffirms the relevance of this type of game-theoretical approach in contemporary conflict analysis.

## Introduction

This article advances conflict studies by applying computational and mathematical modelling, specifically agent-based modelling, to empirically analyse the Kurdish-ISIS conflict in Rojava, Syria. Increasing our understanding of the dynamics of conflict is crucial in evaluating the actions that may lead to their de-escalation and resolution. Unlike previous studies that often focused on identifying causal mechanisms [1, 2] or relied on *artificial societies* [3], this research anchors its analysis in empirical data to dissect the dynamics of strategic interactions and

**Data Availability Statement:** Authors are not authorized to permit, provide, or allow direct access to any of ACLED's original/raw data to any other user due to restrictions imposed by The Armed Conflict Location & Event Data Project

(ACLED). However, interested parties can retrieve data affiliated with this work directly via the ACLED Data Export Tool at https://acleddata.com/data-export-tool/. Queries regarding data can be directed to access@acleddata.com.

**Funding:** This work was partially supported by The Alan Turing Institute (https://www.turing.ac.uk/) under the UK EPSRC grant EP/N510129/1 (A.U. and M.M.) and by The Scientific and Technological Research Institution of Turkey (TUBITAK), ARDEB 1001 Program, Project Number: 120K986 - 'Digital Public Diplomacy of Armed Organizations – Syria and Iraq Cases' (A.U.). The funders had no role in study design, data collection and analysis, decision to publish, or preparation of the manuscript.

**Competing interests:** The authors have declared that no competing interests exist.

violence patterns in conflict scenarios. By integrating game theory into the agent-based framework, we scrutinise the evolving nature of violent events in Rojava, examining whether these incidents follow temporally and spatially stable patterns or vary significantly over time and location. This approach not only fills a notable gap in conflict literature but also provides a sophisticated understanding of the strategic behaviours characterising the Kurdish-ISIS conflict, offering insights into broader conflict dynamics and resolution strategies.

It is crucial to contextualise the study within the broader framework of civil conflict research, a field that, as Kalyvas [1] notes, has gained considerable systematic attention in the last decade, particularly in relation to interstate conflict. The collective classification of civil conflicts, rather than disaggregation into different types of conflict (such as identity and non-identity wars) [4], has resulted in diverging theories that overwhelmingly focus on causes. The economic viewpoint, as suggested by Garfinkel and Skaperdas [5], considers natural resources as a pivotal factor in civil conflict dynamics, a standpoint that Fearon [6] counterargues by suggesting their potential to deter conflict. In international relations, the discourse frequently revolves around ethnic antagonism and pre-existing disputes [7], while comparativists typically emphasise state capacity. Moreover, the dynamics of civil conflict have been extensively explored in terms of civilian violence, with research by Condra and Shapiro [8], Kalyvas [9], and Weinstein [10], among others, examining aspects such as state and insurgent retaliation and the use of violence for recruitment. This multifaceted debate and overlap in civil conflict research provide a critical backdrop for our study's focus on the Kurdish-ISIS conflict in Rojava, Syria, using advanced computational modelling methods.

The extant research on civil wars consistently acknowledges the complexity and fluidity of alliances within such conflicts. Some argue that the strength of alliances varies between civil wars regardless of their nature [11], while Esteban and Ray [12] claim that ethnically homogeneous groups usually present an advantage in forming stronger alliances over others such as class alliances, due to socio-economic heterogeneity. Within this type of conflict, the formation of coalitions is seldom formal, rather these are tacit alliances [13]. In the Syrian context, especially in Rojava, the dynamics of alliance formation among rebel organisations, as explored by scholars like Gade et al. [14], Kausch [15], and Schmidinger [16], are particularly relevant. These alliances, often informal and driven by ideological similarity, are exemplified in the persistent conflict between ISIS and Kurdish militias. Indeed, some of the most intense clashes during the civil war in Syria occurred between ISIS and YPG-affiliated factions, the most intense being the Battle for Rojava and its aftermath. Although much of the international attention on this area was confined to the 2014–15 Siege of Kobani, Kurdish-ISIS conflict ensued much longer than that. This ongoing struggle, intensified by transnational elements such as proxy warfare [17, 18] and the impact of neighbouring conflicts [19, 20], presents a unique case for investigation. The involvement of international actors like the US and Russia, and regional dynamics influenced by the Kurdish People's Protection Units (YPG) and their links to the Kurdistan Worker's Party (PKK), add further complexity. Although our model does not directly incorporate these external influences, their indirect impact on local actors' strategies constitutes important antecedent conditions for our analysis.

In examining the Syrian conflict, particularly in Rojava, Oktav et al.'s [21] analysis of the new generation of violent non-state actors (VNSAs) is pivotal. They classify ISIS as a global revolutionary nonethnic VNSA and the YPG as an ethnic, nationalist VNSA, underlining the distinct identities and global appeal strategies of these groups. ISIS leverages jihadism and the concept of a caliphate for wider reach, while the YPG employs Marxist ideology. Ünver's [22] exploration into the northern region emphasises these groups' capacity to assume state-like functions, as demonstrated by the YPG in Rojava post the expulsion of state forces. Understanding the evolving nature and strategic behaviours of such VNSAs is crucial, and the Rojava

case offers a compelling example of these dynamics. This article extends this understanding by applying advanced computational models to dissect the interactions between these VNSAs in the context of the Syrian conflict.

The study of rebel group dynamics in the Syrian conflict has been a focal point for various scholars. Schwab's [23] research, using data from north-western Syria, introduces a critical dimension of negotiation between rival groups in civil wars, emphasising how the presence or absence of a central authority (like the Syrian regime) influences these interactions. When this conflict is active, rebel groups negotiate as the need for potential allies is higher, whereas in the absence of conflict with the main rival, fighting between rebel groups escalates. These findings are in line with work by Pischedda [24] and Schulhofer-Wohl [25]; the latter looks at on-side fighting in the early years of the Syrian Civil War. Schulhofer-Wohl observes regional variations in on-side violence, with significant instances in regions like Hasaka post-regime withdrawal, contrasting with Damascus where such violence is absent due to ongoing pressure from the regime. This pattern supports Schwab's thesis, highlighting the role of a common enemy in shaping rebel interactions. It is worth noting that Schulhofer-Wohl [25] finds no instances of on-side fighting between the rebel groups we are discussing here, strengthening the case for modelling the Kurdish-ISIS conflict in Rojava. Van Wilgenburg and Fumerton's [26] analysis of the PYD-YPG's consolidation of power in northeast Syria further illustrates the stability of alliances formed against a common rival, leading to the formation of the SDF, dominated by the Kurdish YPG. Their success in power consolidation is attributed to their organisational skills amidst a power vacuum [26, 27]. This article expands on these findings by applying a computational approach to model the Kurdish-ISIS conflict, moving beyond the largely qualitative nature of existing literature.

Computational models dealing with international security have recently emerged within the field of conflict studies (see for example [28]). Statistical models had until now dominated the field [29]; while these are valuable in understanding causal dynamics and the relationship between variables, their application is generally restricted to understanding the causes of conflict [30–32] and forecasting violent outbreaks [33–37]. Some focus in this area has been on understanding the characteristics of a state that may make it more likely to engage in conflict; de Mesquita [38] discusses the application of non-cooperative game theory in conjunction with political economy theory to address this question, and Owsiak and Vasquez [39] (see also [40]) relate this to territorial disputes and the prediction of peaceful dyads. Others have centred on the notion of reciprocity, looking at instances of conflict and cooperation [41–43]. Similar to the work presented in this article, their work seeks to understand strategic response patterns within conflict, particularly relating to reciprocity. However, the focus is on international rather than civil conflict, with actors being aggregated at the country level. Goldstein et al. [41] use conflict event data to show that bilateral reciprocity appears necessary to allow for long-term cooperation, at least in the case of the international dyads they consider in the Middle East. Although these results are derived through a statistical analysis, they exemplify how identifying patterns in conflict event data can lead to a better understanding of conflict dynamics.

In the realm of conflict studies, the last decade has seen a remarkable evolution in the use of computational agent-based models (ABM), which have become instrumental in dissecting the multifaceted dynamics of civil wars. These models, renowned for their capacity to simulate highly detailed interactions among individual actors in conflict environments, have yielded significant insights, contributing to a deeper understanding of emergent conflict behaviours and patterns. The foundational work in this field, although predating the last ten years, is epitomised by Epstein's [44] seminal agent-based model of civil violence which laid the groundwork for subsequent advancements in ABM. This model underscored the influence of varying

perceptions of legitimacy among agents on the escalation or resolution of conflicts. While Epstein's model is the most relevant in this case, others have introduced adaptive behaviour [45, 46] or built models as a tool for policy-making rather than theory testing [47]. Empirically-driven models have also emerged in the wider field of conflict studies. Bhavnani et al.'s ground-breaking 2014 study [48] ventured into urban violence within ethnically divided cities. By integrating geographical and social network factors, their model offered a nuanced perspective on urban settings' role in conflict dynamics.

In 2021, Mueller [49] expanded the scope of ABM to ethnic violence. His work incorporated critical variables like ethnic heterogeneity and political exclusion, offering insights into the complex interplay of ethnic tensions and political structures in conflict initiation and sustenance. Studies such as Schutte and Donnay's [50] marked a significant shift toward forecasting violent hotspots. Their integration of geographic and temporal data into ABM provided a sophisticated understanding of the evolution of conflict dynamics.

Further pivotal contributions came from Cederman, Gleditsch, and Buhaug [51], who used ABM to investigate the links between political and economic inequalities and conflict. Their focus on African ethnic conflicts provided a rich context for examining how resource allocation and ethnic disparities could catalyse and propel conflicts. These models have been applied to various contexts, with a significant emphasis on the Syrian Civil War and African ethnic conflicts, although this specialised field is very much in its first stages and will undoubtedly experience significant advances in the coming years.

A cornerstone in the evolution of ABM is Axelrod and Bennett's Landscape Theory of Aggregation [52]. Although their work dates back to 1993, it laid the conceptual groundwork for understanding how local interactions could escalate into larger conflict patterns, a principle that has been integral to later developments in ABM. Another significant stride in ABM research is seen in the work of Turchin et. al. [53], who have delved into the socio-political underpinnings of civil unrest. Their models have been particularly adept at simulating the complex social dynamics that lead to civil uprisings, offering insights into the threshold points of collective action and rebellion. Further, the work of Axtell, Epstein, and Young [54] on network dynamics in conflict zones has provided a fresh perspective on the role of social networks in conflict evolution. The incorporation of geographical and socio-economic data in ABMs, as seen in the work of Weidmann and Salehyan [55], has also been critical. Their models have been pivotal in understanding how geographic and economic factors intertwine with political dynamics to shape the landscape of civil conflicts. In addition, the recent shift towards integrating real-time data into ABMs, exemplified by the work of Weidmann and Cederman [56], has brought an unprecedented level of accuracy and relevance to conflict simulations. This approach allows for a more immediate understanding of ongoing conflicts, providing insights that are crucial for timely policy interventions. Collectively, these advancements in computational agent-based modelling have profoundly transformed the landscape of conflict studies. By encapsulating the multifaceted interactions of diverse agents and incorporating a wide range of variables, from geographical to socio-political factors, these models have provided comprehensive insights into the dynamics of civil wars and potential pathways to resolution.

A significant trend in the last five years is the integration of real-time data into ABMs; a methodological advancement that has been crucial for analysing ongoing conflicts with greater accuracy and relevance. For instance, in the study of the Syrian Civil War, researchers have utilised ABMs to dissect the complex interplay between various factions, as seen in the works of Moro [57] and Suleimanova et. al. [58] who applied real-time data to model the shifting alliances and territorial control within the conflict. This approach has been instrumental in providing timely insights for policy interventions. The integration of geographic and socio-economic data into ABMs has also been prominent, particularly in studies focusing on African

civil wars. For example, Bruch and Atwell [59] research incorporated spatial and economic variables to understand how geographic factors and resource allocation influence political conflicts. Their study highlighted the intricate relationship between land, resources, and conflict, offering a multi-dimensional view political conflicts. Network dynamics in conflict zones have been another focal area in recent ABM research. The study by Fjelde and Hultman [60] on the role of social networks in the escalation and de-escalation of conflicts provided valuable insights into how information flow and social influences shape conflict dynamics. Their model emphasised the impact of inter- and intra-group communications in conflict zones, offering a nuanced understanding of the social underpinnings of civil wars.

This article represents an advancement in the field of conflict studies, particularly in the application of agent-based models and game theory to the analysis of violent conflict and civil wars. It offers a distinct improvement over previous scholarship by adopting a data-driven approach rather than relying on artificial societies, utilising the micro-level event dataset from the Armed Conflict Location and Event Data Project (ACLED) to model the behaviour of non-state actors, specifically ISIS and Kurdish militias, in Rojava from January 1, 2017, to December 31, 2019. This study diverges from traditional methodologies that predominantly focus on state military dynamics, instead delving into the less-explored territory of dyadic interactions between two non-state actors. It specifically addresses the conflict between an ethno-nationalist armed group (Kurdish militias) and a religious fundamentalist armed group (ISIS), a type of interaction that has not been extensively covered in the existing literature. This nuanced focus allows for a deeper understanding of the strategic behaviours and interactions between groups with different ideologies and agendas in such conflicts. Methodologically, the study employs an Iterated Prisoner's Dilemma game, also represented as an instance of the Hawk-Dove game, to model the conflict, offering a more robust framework for analysing the strategic decisions of the involved parties. This approach reveals that shifts in territorial control are more influential than geographical or temporal variables in determining patterns of violent action. It also highlights that periods of one-sided violence and the proportion of attacks by each side are more indicative of significant changes in the conflict than previously recognised patterns of strategic behaviour. The choice of the Kurdish-ISIS conflict in Rojava is particularly significant due to its dyadic and multi-ideological nature [16], allowing for a clearer modelling of interactions between the two non-state actors with different non-rational priorities in conflict. We also emphasise this ideological component in our analysis, and the bulk of the agent-based modelling literature has so far treated all conflict actors as having uniform priorities and strategies, which is rarely the case in a conflict environment.

This focus also acknowledges the strategic importance of Rojava; traditional game theoretic studies would struggle to model the importance of this area as it lacks natural resources or a specific strategic importance such as serving as a natural barrier or a passageway between key logistical hubs. Instead, Rojava is important due to its ideological implications, as it is viewed among the Kurdish rebel groups as the home of transnational Kurdish revolutionary ideology [61, 62]. It is important to note that this dyadic interaction forms part of a wider conflict involving additional actors, both on a national level and external actors. While Kurdish and ISIS forces have also interacted with other actors in the wider conflict, the motivation for choosing the Rojava region in particular is that this has predominantly seen conflict involving the Kurdish-ISIS dyad. Nevertheless, the implications of focusing on this interaction in isolation is discussed below.

The study's contributions are threefold. It not only employs a classical game theory-based approach to analyse a unique and novel conflict with highly novel forms of actor behaviour and ideologies, but it also makes a theoretical contribution by examining the factors that influence such strategic behaviour. Moreover, the use of micro-level data to investigate the Rojava

conflict represents a departure from the conventional approach of modelling artificial societies with meso-level data, providing a more granular and dense analysis of conflict dynamics. In summary, this research significantly enriches the current understanding of conflict dynamics, especially in the context of non-state actors, by combining empirical data analysis with advanced game-theoretic modelling, offering fresh perspectives and methodologies to the field of conflict studies.

## Materials and methods

### Conflict datasets

The choice of dataset was crucial as modelling actors' behavior required sufficient information per event recorded as well as a large number of events in order to derive meaningful results. The recent surge of interest in the microanalysis of civil conflict has given rise to a number of event data projects [63], in the hope that disaggregation will allow for a better understanding of causal mechanisms [64, 65]. Some of the more widely used datasets include the UCPD/ PRIO Armed Conflict Dataset (see [66, 67]), the Armed Conflict Location and Event Data Project (ACLED) (see [68, 69]), and Integrated Conflict Early Warning System (ICEWS) (see [33]). These datasets all rely largely on news coverage to document politically sensitive events, meaning that an important consideration in selecting the appropriate dataset for this analysis was the mitigation of bias. Considering this and other factors including time-period and the level of detail within the data, ACLED was deemed the best choice for this paper, acknowledging that shortcomings remain. Details on data extraction and pre-processing are included in S1 Appendix.

Given that the data covered relates to conflict, even a wide variety of sources cannot entirely remove the presence of data bias due to the political nature of events [65], the inherent difficulty in recording accurate data in conflict [35, 70, 71], and the fact that news reports often cater to a domestic audience and have a political agenda [72]. Regarding estimates of the number of people killed, Price, Gohdes and Ball [71] point out that existing data constitutes a convenience sample rather than an accurate representation of the number of people killed in the conflict. Although here we model conflict events rather than individual battle deaths, the question of under-reporting remains a central matter. ACLED have attempted to mitigate these issues by using non-governmental sources with no direct political involvement and a combination of local, national and international sources [73], which has been shown to reduce data bias [74], pertinently urban bias in civil war [75].

Although a certain level of urban bias remains in the ACLED dataset and there is some lack of precision in geocoding [76], the level of precision is sufficient for this research. Most other conflict datasets rely on similar news outlets; consequently, the effectiveness of cross-checking is limited. The UCDP/PRIO dataset, widely used in conflict literature, was considered as an alternative. However, their data for Syria lacks disaggregation and presents some inconsistencies [77]; its lower quality is precisely the reason why Syria's data is excluded from the main UCDP/PRIO dataset. Given the absence of alternative datasets, the presence of bias was mitigated to some extent by cross-checking the results with news coverage, primarily using The Carter Center's [78] weekly conflict reports. From this comparison, there appeared to be some underreporting in the ACLED dataset, more marked for ISIS attacks. Nevertheless, all battles and offensives were reflected in the dataset, only some smaller-scale attacks were missing. While this highlights limitations in the data, it should not have a major impact on the modelled dynamics.

Weidmann [79] suggests methods to counteract reporting bias, such as capture-recapture methods or sensitivity analysis, however these either apply to statistical analyses or are outside

the scope of this paper—he nevertheless concludes that there are currently no effective solutions. Therefore, following the approach taken by a number of studies (see [32, 80]), the assumption is that bias is inevitable, yet it is not strong enough to invalidate conclusions.

## Modelling approach

In order to understand how the behaviour of actors in the Kurdish-ISIS conflict in Rojava changed both geographically and temporally, their behaviour was first modelled as an iterated "Prisoner's Dilemma" game, also represented as an instance of the Hawk-Dove game. Axelrod [81] highlighted the appropriateness of this game as a model for issues in international politics, particularly those lacking a central authority, as it applies to situations where self-interested behaviour leads to a worse outcome for both actors. There are numerous examples of the use of this framework to model interactions within conflict studies literature, both focusing on individual conflicts [82, 83] or from a theoretical perspective [84, 85]; other approaches have used alternative game specifications [86, 87]. In the Prisoner's Dilemma, two players must simultaneously choose between cooperation or defection. In a one-shot game, the incentive to defect is higher, although mutual defection is suboptimal; in an Iterated Prisoner's Dilemma, mutual cooperation is beneficial if there is an expectation of reciprocation from the opponent. In this case, we are applying the Iterated Prisoner's Dilemma, so players' actions may be dependent on the past actions of their opponent; the two players we model are ISIS and Kurdish militias.

The Hawk-Dove model [88] was originally designed to represent interactions within animal behaviour; however, it has since been integrated into conflict studies and wider international relations and political science literature. Its use within this literature has often become decoupled from the original formulation, resulting in the use of *Hawk* and *Dove* as ways to describe behaviour rather than specific actions in the original game (see for example [89–93]). The model involves two parameters: the value $V$ of a resource and the cost $C$ of fighting over this resource. In the case where $C \leq V$, the Hawk-Dove model becomes identical to the Prisoner's Dilemma. A *Dove* and *Hawk* in the former map directly onto *Cooperator* and *Defector* strategies in the latter. For the following analysis, we will continue to use the vocabulary from the Prisoner's Dilemma; however, due to the widespread use of the Hawk-Dove model, it is important to note that the research presented in this article can be understood through the lens of both frameworks.

Employing the Iterated Prisoner's Dilemma and the Hawk-Dove game to study the conflict behaviour of ISIS and the Kurdish YPG is particularly effective due to the distinct ideologies and objectives of these two organisations. These classical game forms offer a structured and clear framework for analysing the strategic decision-making processes in scenarios where actors face choices between *fighting* and *doing nothing*, which is crucial in understanding the dynamics between these ideologically divergent groups. In the context of ISIS and the Kurdish YPG, the Prisoner's Dilemma provides a profound insight into situations where both parties face the choice between aggressive actions and restraint. The model is apt for illustrating the dilemma faced by ISIS, with its religious fundamentalist objectives, and the Kurdish YPG, driven by ethno-nationalist goals, in determining which areas they should attack, which ones they should defend, and which others they should relinquish (*do nothing*).

Due to the impossibility of cooperative options in this scenario (such as trade, contextually ally, or conduct joint peacekeeping efforts) given to the nature of the strategic interaction (non-converging ideologies), the inherent distrust and conflicting objectives push them towards non-cooperative strategies. This dynamic is central to understanding the ongoing conflict, where mutual best interests are overshadowed by the drive to protect individual

ideological goals. The Hawk-Dove game, with its focus on confrontational versus restraint-oriented strategies, further enriches this analysis. This game is particularly relevant in examining scenarios where one group opts for aggressive tactics while the other chooses a more defensive or non-intervening stance, effectively capturing the asymmetry in the conflict. The divergent ideologies and goals mean that the payoffs for aggression and conciliation differ for ISIS and the Kurdish YPG, influencing their strategic choices in a complex interplay of power and diplomacy.

The simplicity and clarity of the Iterated Prisoner's Dilemma and Hawk-Dove games, in contrast to more complex and variable-rich models, offer distinct advantages. They provide a focused analysis of critical decision points in the conflict, directly addressing the strategic interactions between the two parties. This direct focus is essential in a situation where each group's actions significantly influence the other's strategies. Moreover, these models adeptly model the ideological contrasts between the groups and are effective in incorporating the historical context of the conflict, offering insights into how past actions influence current strategic decisions. Despite their simplicity, these models have a predictive power that is crucial for understanding potential future developments in the conflict. In summary, while newer agent-based models and advanced game-theoretic systems might offer more complexity, the classical forms of the Iterated Prisoner's Dilemma and the Hawk-Dove game provide a focused, strategic lens through which to view the conflict between ISIS and the Kurdish YPG. Their ability to distil the essence of strategic interaction, especially in the context of differing ideologies and objectives, makes them highly suitable and effective for this particular case study.

In our analysis, we map patterns of *cooperation (Dove)* and *defection (Hawk)* from playing strategies against each other to observed actions recorded in the ACLED dataset. Although in the Prisoner's Dilemma model the two actions are denoted as *cooperation* and *defection*, it is important to note that we are not interpreting the lack of violent action observed in the ACLED data as instances of 'cooperation' in the conflict itself. In using conflict data, only *defection* in the Prisoners' Dilemma game is visible; we are thus using the Prisoner's Dilemma as a framework to model *defection* and *inaction*, where the latter is taken to mean a lack of reported violent action; in the context of our model based on the Prisoner's Dilemma game, inaction can be considered as a form of non-defection. Therefore, strategies that frequently resulted in sustained mutual cooperation in an Iterated Prisoner's Dilemma were discarded, as these would represent a lack of action from either actor in the conflict and as such would not be observed in the data. In particular, the patterns of actions were mapped to the strategies submitted to Axelrod's first and second tournaments [81] along with other types that have emerged in later literature, including zero-sum strategies [94] (S2 Appendix, S1 and S2 Tables). Probabilistic strategies [95] were excluded as they did not produce repeating patterns, therefore a characteristic sequence of actions could not be extracted. Strategies covered in this analysis include, for example, *Tit-for-Tat* (TFT), where a player begins by cooperating, then copies the opponent's last move and *Alternator* (ALT), where a player alternates between cooperating and defecting each round.

Having selected these strategies, each one was played against all others and itself using the Axelrod Library on Python [96] (see S2 Appendix), and the resulting patterns of defection were recorded. For instance, if both players are following *TFT*, no defection would be observed as both players would cooperate in each round. Alternatively, if one player is employing *TFT* and the other *ALT*, we would observe alternating defections from each player. From this, a pattern of defection by two players can be matched to one or more strategy combinations. To apply this to the ACLED data, violent actions by one side of the conflict or the other were first converted into an ordered sequence in a string format. Taking all the possible patterns of defection, we then extracted the number of occurrences of each of these patterns within the

resulting strings. From this, we could identify the most frequent interaction between the two actors for that subset of data. Using the strategy categorisation below, we can map which combination would lead to such an interaction (e.g. if we observe a pattern of alternating defections, this could occur from *TFT* and *ALT* strategies being played). We can thus identify the strategic interaction this corresponds to, and by deriving the behaviour of each actor we can analyse behavioural shifts throughout the conflict.

This method gave rise to a potential source of uncertainty: the most frequently occurring patterns returned by the model may in fact occur as a subpattern of another made up of a higher number of characters. For example, from a pattern of two defections by Player *A* followed by a defection by Player *B*, we would conclude that possible strategic interactions are *DEF* vs. *FBF* or other such combinations that result in this pattern. However, the observed pattern may occur as a subset of two defections by Player *A* followed by two defections by Player *B*, for which the correct interactions include *BU* vs. *TFT* or *DES* vs. *DDC* (for detailed strategy categorisation, see S3 Table). This was therefore taken into account in the analysis: where shorter patterns occurred, we verified what longer patterns these could be part of, as well as whether those longer patterns were observed in the data. Both possible combinations of strategies were then considered. In our model, Player *A* represents Kurdish militias and Player *B* represents ISIS.

## Data selection

In the selection of the geographical and temporal subset of event data to model, it was paramount to have a representative range of variation in time, location and territorial control. Without this, any changes in strategy cannot decisively be attributed to any single one of these factors. The locations with the highest number of recorded events in the ACLED dataset were chosen, as bias or recording errors should be minimised in those with a large amount of data [64]. Table 1 details the 13 locations with 50 or more events for these actors in the Rojava region that are the starting point for the analysis. Raqqa (or Ar-Raqqa), for example, is the city with the highest number of recorded events, at 1,314 out of a total 6,061. This is unsurprising considering the battle of Raqqa, 6 June–17 October 2017, has been one of the most violent events of the conflict [97].

**Table 1. Modelled locations.** Summary of the locations modelled in this analysis; these are all the locations with 50 or more events recorded for ISIS and Kurdish forces in the Rojava within the period 1 January 2017–31 December 2019.

| Location | Urban/Rural | Territorial Control | Recorded Events |
|---|---|---|---|
| Raqqa | Urban | ISIS → Kurdish forces | 1,314 |
| Al Bukamal | Urban | ISIS → pro-government forces | 574 |
| Deir ez-Zor | Urban | Contested → pro-government forces | 559 |
| Hajin | Urban | ISIS → Kurdish forces | 285 |
| Al Tabqa | Urban | ISIS → Kurdish forces | 206 |
| Shadadah | Urban | Kurdish forces | 205 |
| Bahguz | Rural | ISIS → Kurdish forces | 131 |
| Sosa | Rural | ISIS → Kurdish forces | 110 |
| Shafa | Rural | ISIS → Kurdish forces | 99 |
| Al Bab | Urban | ISIS → Rebel forces | 73 |
| Al Mayadin | Urban | ISIS → pro-government forces | 64 |
| Al Hassakeh | Urban | Contested | 60 |
| Basira | Urban | ISIS → Kurdish forces | 50 |

The dataset was broken down into three-month segments for each location to ensure temporal accuracy in the analysis. The unit of analysis for this research was thus the conflict between ISIS and Kurdish militias over the period 1 January 2017–31 December 2019, modelling individual locations over three-month segments. The locations are modelled individually in order to identify local interactions, however when analysing the results we also take these interactions in the context of the wider conflict, and not as isolated battles or events. Due to the Covid-19 pandemic, events after 31 December 2019 were not considered as we expect a disruption in the recording of events.

As mentioned above, the ACLED dataset does contain a certain amount of urban bias [76] —this is relevant in our case, as ten out of the thirteen locations modelled were urban areas (Table 1). However, events with a higher number of observers tend to have lower reporting inaccuracies [65], indicating that data from urban centres may be of higher quality. Moreover, locations with fewer recorded events are more likely to be underreported on, thus omitting relevant events. This would lead to the incorrect coding of certain periods as peaceful, leading to a measurement error [79]. Having said this, the comparison with news coverage mentioned above did not reveal any significant underreporting in rural areas.

Most of the modelled locations are on the banks of the Euphrates. This is not surprising, as much of Rojava's strategic importance lies in its wealth in water [98], leading to the banks of the river having become a *locus* of the fighting. Nevertheless, these locations do present geographical diversity and thus allow for variation in the analysis.

As the event data records all political conflict, numerous actors that are not being considered in this research were recorded, including some categorised as *Unidentified Military Forces*. Since we focused on the Kurdish-ISIS conflict, these were discarded when it came to modelling the data.

## Results

### Strategy categorisation

In order to identify the strategic interactions between actors in the Kurdish-ISIS conflict in Rojava, each strategy was played against all others and itself (S1 and S2 Tables, S2 Appendix). The shortest repeated pattern of minimum length 3 for the resulting interactions were first categorised (S3 Table). For instance, playing *TFT* against *ALT* would result in a pattern of alternating defection, whereas the outcome of *ALT* against *DEF* would be a defection by Player *A* for every two by Player *B*. These interactions are visualised in Fig 1, to show the strategy combinations that result from more *offensive* or *defensive* actors: taking Player *A*, a blue square represents a defensive strategy as all defections/attacks are carried out by Player *B*. For instance, *DEF* and *HTFT* are generally offensive strategies no matter what they are playing against; conversely, *COP*, *TF2T* and *WL* are generally defensive strategies.

The resulting patterns of defection were then mapped onto the event data for Rojava through a method of string-matching to identify recurring strategic interactions. That is to say, we categorised the resulting patterns of defection from playing these strategies against each other, then identified which of these patterns were present in the events dataset. Player *A* and *B* were substituted for actors in the conflict, allowing us to track strategic interactions recorded in the ACLED event data. Several strategy combinations may result in the same pattern of defection; it is therefore not possible to identify distinct strategies using ACLED data. Instead, we identify the combinations of strategies that could result in a given observed emerging pattern. We take Player A to be Kurdish forces, and Player B as ISIS; we may observe for instance a pattern where every three attacks by Kurdish forces in a certain location are followed by an ISIS attack, represented below as 3:1 (Kurdish). Note that this does not refer only

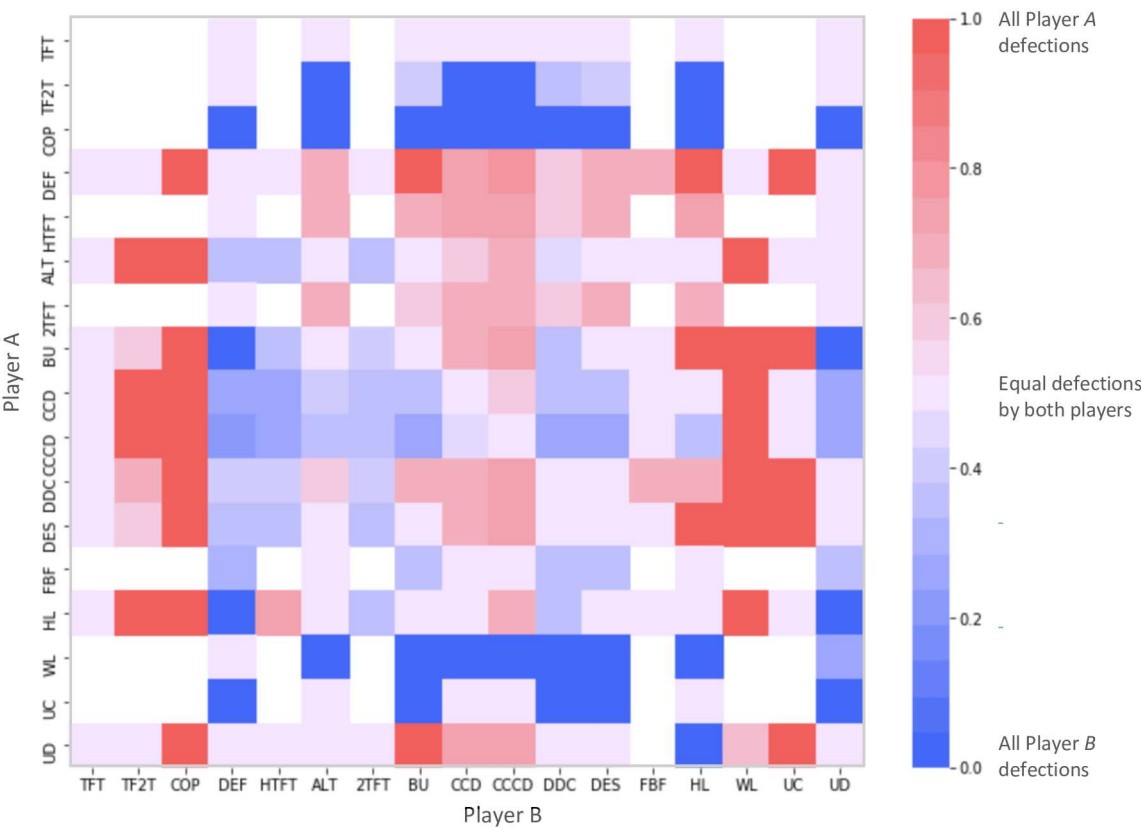

**Fig 1. Visualised strategy categorisation.** Illustrates the proportion of attacks carried out by Player *A*. A red square represents an offensive strategy as all defections/attacks are carried out by Player *A*, whereas a blue square represents a defensive strategy as all defections/attacks are carried out by Player *B*. A white square denoted sustained cooperation from both players.

to the proportion of attacks carried out by each side, specifically that every three Kurdish attacks are followed by an ISIS attack. This pattern in particular was observed in Raqqa, particularly over the period January–September 2017 and again towards the end of 2019. As mentioned, the observed pattern may be the result of a number of strategy combinations. A pattern of three *defections* by Player A followed by one by Player B could be the outcome of combinations such as *DEF* vs. *CCD* or *HTFT* vs. *HL*.

## Modelling results

Locations with 50 or more events recorded for ISIS and Kurdish forces in the Rojava region were modelled—these locations are presented in Table 1, showing the wide variation in the number of events recorded. This variation restricted the choice and thus the geographical variation in the model. For those with fewer events, it was not possible to model the strategies for each segment of time, placing further constraints on the conclusions that can assertively be drawn from the data. While it is clear that some locations have experienced a higher number of events compared to others, the numbers below likely do not accurately reflect the total number of violent events in each location.

ACLED categorises entries into *Battles, Explosions/Remote violence, Protests, Riots, Strategic Developments* and *Violence Against Civilians*. Only those pertaining to conflict were relevant to this paper, in particular *Battles* and *Explosions/Remote violence*. They are defined

respectively as "violent interactions between two organised armed groups" and "one-sided violence events in which the tool for engaging in conflict creates asymmetry by taking away the ability of the target to respond" [99]. These two event types were first mapped to identify any geographical clustering; both display a relatively homogeneous spread of events. Events classed under *Violence Against Civilians* were excluded as we cannot assertively situate these events within the present conflict; however, in so doing some relevant events may have been excluded, both because they may have been misclassified and in reality did involve the actors we are studying, or because this was a strategic action by one of the actors as part of the Rojava conflict.

The count of events did show some variation between event types. While in both cases more events originated from Kurdish forces, the difference is much more marked for battles. 12.4% of battle events have ISIS as the causal actor, compared to 28.3% for remote violence. However, the significantly higher number of events related to Kurdish forces may to some extent be a symptom of underreporting being more marked for ISIS events, as indicated following comparison to news coverage.

Aside from data bias, one of the major limitations of our analysis is the lack of allowance for transnational actors and influence. Although external actors have been heavily involved in the Rojava conflict, this involvement has often been indirect and, therefore, is not reflected in the data. While this does not mean that strategic variations are incorrect, it could lead to external influences not being identified if they are not well documented. Due to these events' political nature, it is to be expected that a large part of these developments, including arms sales or indirect support, will not be publicly covered. Therefore, while attempts have been made to account for the transnational dynamics of the conflict, some of these impacts will unavoidably be present in the results.

## Discussion

In analysing the results of the model, three main factors were considered: temporal variation, geographical variation, and changes in territorial control. Below, the impact of each of these on the behaviour of actors in the Kurdish-ISIS conflict in Rojava in the period 1 January 2017–31 December 2019 is considered, modelled per location and three-month segment.

### Geographical variation

The results show some geographical consistency—strategic patterns remain relatively constant within each location. An example is *Al-Bukamal*, where at least 83% of all interactions per time period consist of the same observed pattern of one-sided violence where all attacks are carried out by Kurdish forces (this can be derived from a number of strategy combinations but require Player *B* (ISIS) to display no retaliation). The most frequently occurring pattern in all but two locations (Raqqa and Basira) is again one-sided attacks by Kurdish forces, although we do observe some attacks being carried out by ISIS—in this case every four Kurdish attacks are followed by an ISIS one, denoted as 4:1 (Kurdish) (Fig 2—note that proportions add up to more than 1 as one pattern may occur as a subpattern of another). From the strategy categorisation above, the only strategy combination that would result in this pattern is *DEF* played against *CCCD*. The reverse can be seen in Basira, where we observe four recorded ISIS conflict events for every one from the Kurdish side; this pattern would result from *CCCD* played against *DEF*. Notably, neither of these strategies take into account the other player's actions in determining their own behaviour—the frequency of their defection is constant regardless of their opponent's behaviour. This may point to attacks being carried out less as a reactionary move and more as part of an overarching strategy.

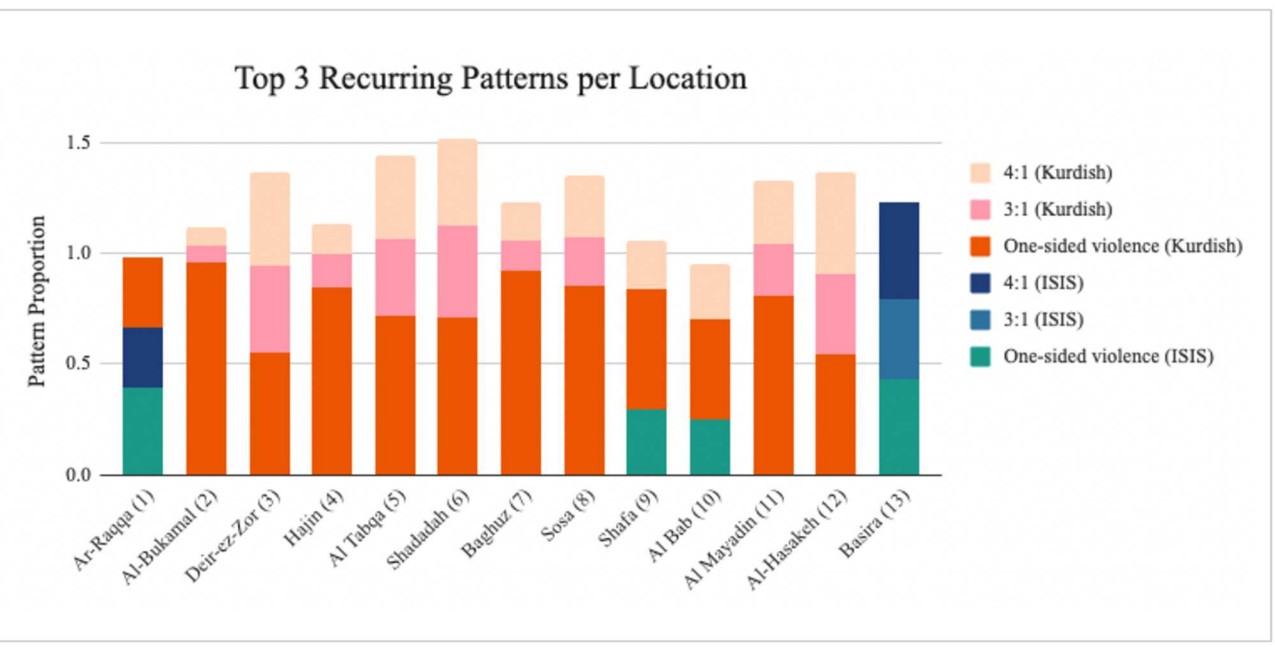

**Fig 2. Recurring patterns per location.** Data source: [100]. *4:1 (Kurdish)* denotes that the recurring pattern observed consists of four conflict events carried out by Kurdish forces for every one carried out by ISIS. One-sided violence corresponds to the repeated pattern of shortest length three of attacks carried out by the same side. For detailed strategy categorisation, see S1 Table.

Punctuated violent outbursts occur in a number of locations, including *Sosa* and *Al-Maya-din*, where other periods lack sufficient data for the behaviour to be modelled. However, those phases that we can model present homogeneity in terms of observed patterns, indicating geographical consistency in strategies.

Contextualising each location within the conflict is essential; individual battles are often part of a larger offensive and thus cannot be taken in isolation. Kurdish forces pushing the ISIS frontline gradually further south [101] accounts for the geographical clustering of modelled locations along the banks of the Euphrates—the movement of troops would plausibly lead to similar strategic interactions in these locations sequentially. The data reflects precisely this phenomenon: the observed patterns are very similar across these locations, primarily made up of short stints of one-sided violence, but they occur at different times. As Kurdish militias advance southwards towards Bahguz, a surge in violence can be observed in *Hajin* from early 2018. This is accompanied by smaller outbursts first in *Shafa* and then *Sosa* and *Bahguz* towards the end of 2018 and beginning of 2019 (Fig 3). We then cannot identify geography as the causal factor in this case, as the movement of troops is crucial to these developments. Changing territorial control and moving frontlines are both dictated by geography and more accurately account for the outbursts of violence.

## Temporal variation

Temporal variation presents similar limitations to geography; in this case constraints are more marked as there are insufficient events to compare each segment across all locations. The time variable appears less consistent—the strategic interactions do not change over time in the same way across different locations. *Raqqa* presents a complete reversal of strategies between the two sides within the period studied (Fig 4), although the number of recorded events

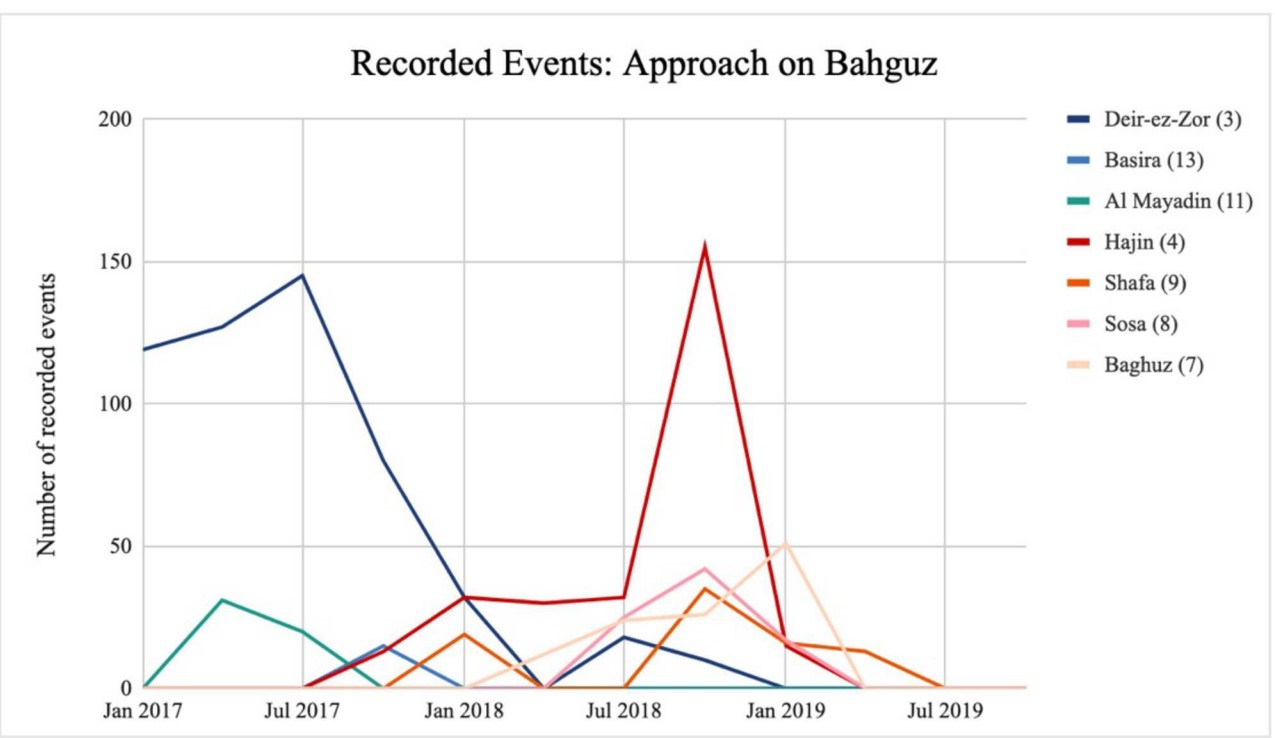

**Fig 3. Recorded events: Approach on Bahguz.** Data source: [100]. Number of recorded events per location in the time preceding the approach on Bahguz in early 2019. In most of these locations, we observe a change in territorial control from ISIS to Kurdish forces.

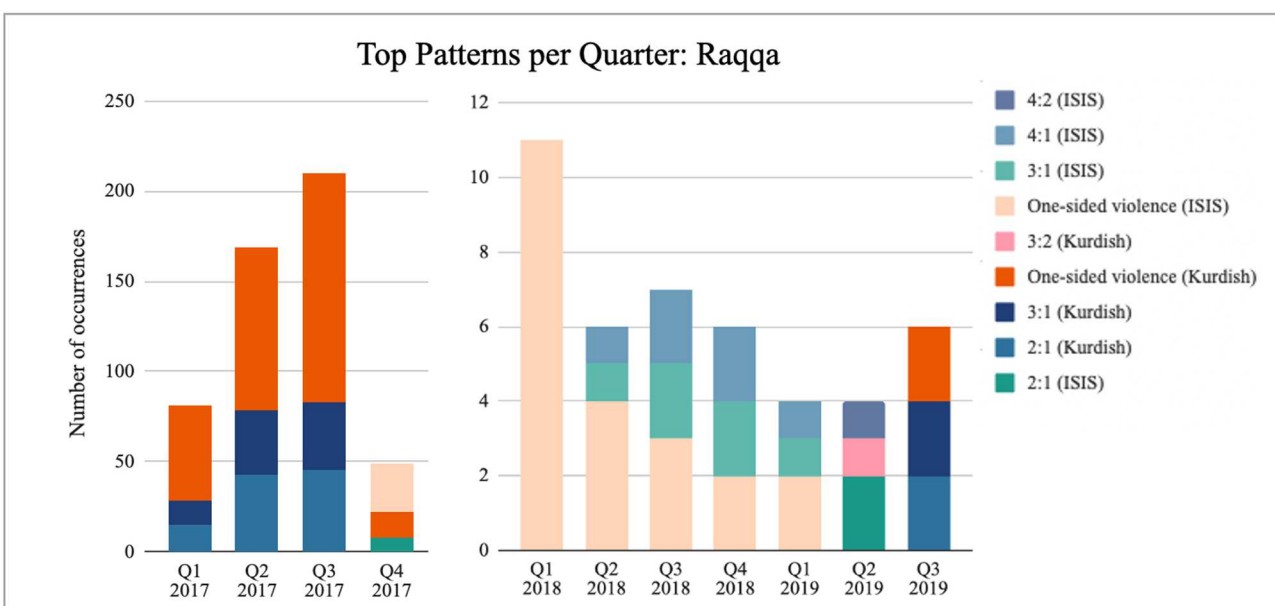

**Fig 4. Top 3 patterns per quarter: Raqqa.** Data source: [100]. Depicts how many times we observe the three most frequently occurring pattern for each quarter in Raqqa. *4:2 (ISIS)* denotes that the recurring pattern observed consists of four conflict events carried out by ISIS for every two carried out by Kurdish forces. One-sided violence corresponds to the repeated pattern of shortest length three of attacks carried out by the same side. For detailed strategy categorisation, see S1 Table.

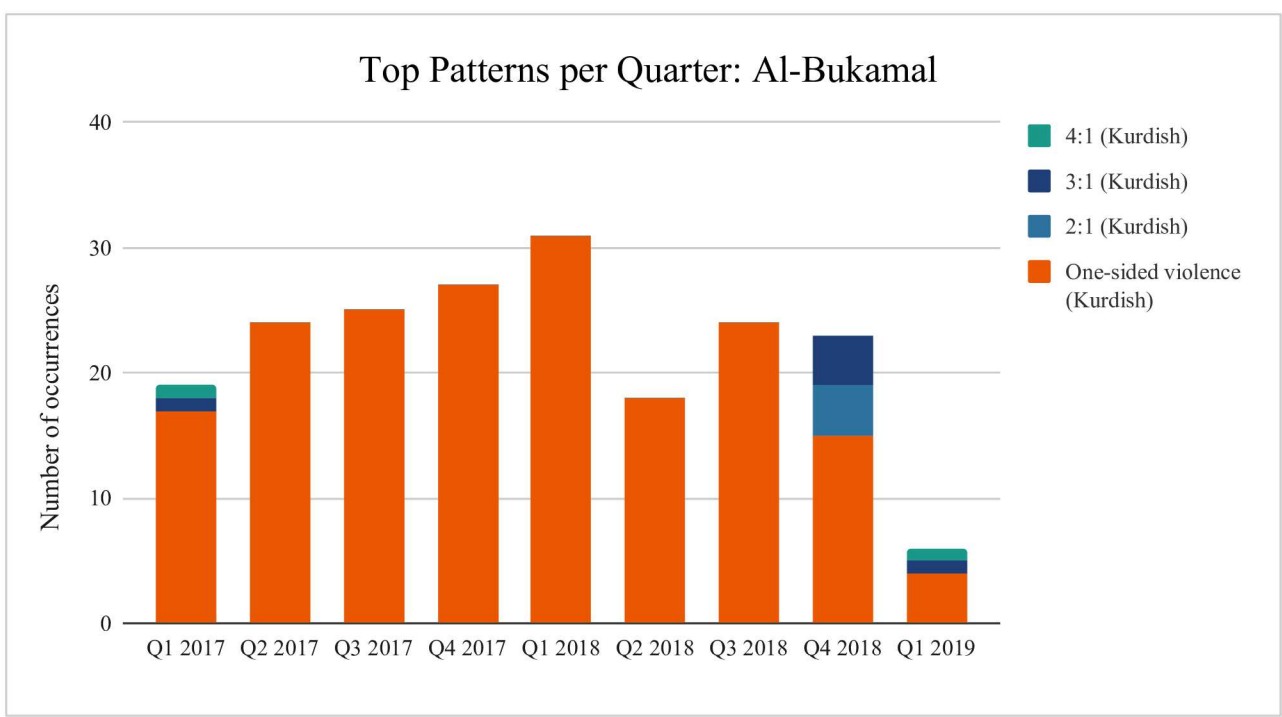

**Fig 5. Top 3 patterns per quarter: Al-Bukamal.** Data source: [100]. Depicts how many times we observe the three most frequently occurring pattern for each quarter in Al-Bukamal. *4:1 (Kurdish)* denotes that the recurring pattern observed consists of four conflict events carried out by Kurdish forces for every one carried out by ISIS. One-sided violence corresponds to the repeated pattern of shortest length three of attacks carried out by the same side. For detailed strategy categorisation, see S1 Table.

significantly reduces from late 2017 until the end of the period. Both the reversal of strategies and the reduction in events coincide with the end of the Battle of Raqqa [97]. *Shafa* presents a similar case, although with considerably less data available. Conversely, the strategic interactions in *Al-Bukamal* appear to be unaffected by temporal variation (Fig 5). As with geographic variation, this indicates that another variable is at the root of the changes we observe, namely territorial control. Looking at control shifts in Syria over this time period, we can see that passing time directly reflects the loss of territory by ISIS compared to gains on the Kurdish side [102].

It is not surprising that we see no direct relationship between temporal variation and changes in strategy or observed patterns of interaction, considering no particular stages or clear dynamics have been identified within civil conflict literature. In the Rojava conflict, particularly in the period 2017–2019, the shift of territorial control was primarily in one direction and relatively constant. Therefore, as in this case time is closely related to territorial power shifts, we cannot draw general conclusions from the present model as there is insufficient variation within the data analysed.

## Territorial control

Single locations cannot be taken in isolation if we are to understand how this type of modelling can be used to comprehend and potentially forecast changing dynamics. By considering the shifts in territorial control in surrounding areas, we obtain a much clearer idea of how strategic interactions may change in the future. Comparing violent outbursts observed in the modelled

data to maps of the changing power structure show that the movement of troops usually goes hand in hand with sequential outbursts of one-sided violence, primarily carried out by said troops. For example, as mentioned, the advance of Kurdish troops mirrors the sequential surges of violence in *Shafa*, *Sosa* and *Bahguz* (shown by the prevalence of the pattern of Kurdish attacks met with no retaliation, and occasionally the contrary)/

The type of actor did not appear to dictate the observed behaviour. As we have seen, a pattern of one-sided Kurdish attacks met with no retaliation can emerge from a number of strategy combinations, although Player *B* (ISIS) must be following one of the following strategies: *TF2T, COP, HL, WL, UC, BU* (vice versa for periods of one-sided ISIS attacks). Aside from *COP*, where Player *B* always cooperates regardless of the other's actions, the rest are reactive strategies that respond to the opponent's prior actions. An actor following the strategy *HL* will defect only to mutual cooperation, conversely *WL* employs defection only following mutual defection. Greater data availability could allow for the identification of the individual strategies being followed. In particular, having data for events relating to cooperation would allow for the modelling of both actions in an Iterated Prisoner's Dilemma, and so a more accurate identification of the strategy combinations. In turn, this could be used to understand what actions might contribute to de-escalation and conflict resolution.

It must be noted that by considering the data in hindsight we run the risk of confirmation bias. Territorial control allows us to estimate an area where an ensuing violent outburst may occur, not necessarily the precise location. A number of other factors need to be taken into account in order to make a more accurate prediction, including the strategic value of surrounding locations, and the positioning of troops. As the period 2017–2019 shows a relatively constant gain in territory by Kurdish forces, it does not allow for the consideration of how strategies would vary was the power balance reversed.

Confirmation bias must also be considered in terms of data reliability. The ACLED dataset is built on news coverage, meaning that events with significant results will have wider coverage, indicating that power shifts may in reality derive more retaliation than suggested by this model as instances where this retaliation fails will garner less news coverage. Whereas individual locations do not display retaliation by ISIS, taking the data as a whole reveals that the proportion of attacks carried out by ISIS increased as they lost control of territory, particularly in 2019 (Fig 6). Therefore, by taking the conflict as a whole it is clear that a shifting power structure alters the strategic interactions between the two sides. The fact that we do not observe this for individual locations may be due to the number of retaliatory events per location being too small to be significant when modelling the data, or to the fact that retaliation may not always take place in the location where attacks or territorial takeover happened. It must be noted that reporting bias may account for some of this increased recording of events. Price, Gohdes and Ball [71] discuss how control over territory impacts the level of documentation of deaths in particular as human rights documentation groups are freer to work in areas under the control of one side of the conflict. Regarding the Syrian case, they "suspect that the relatively lower number of killings reported in regions under Daesh control reflect changes in documentation dynamics, not in conflict dynamics" [71]. The fact that we observe a greater number of ISIS attacks and retaliation as Kurdish forces gain control of more territory may then in part be a result of greater reporting in those areas.

## Conclusion

Our model reveals that the proportion of attacks carried out by each side is a more indicative measure of significant changes in the conflict than previously recognised patterns of strategic interaction. This finding challenges conventional approaches in conflict analysis, which often

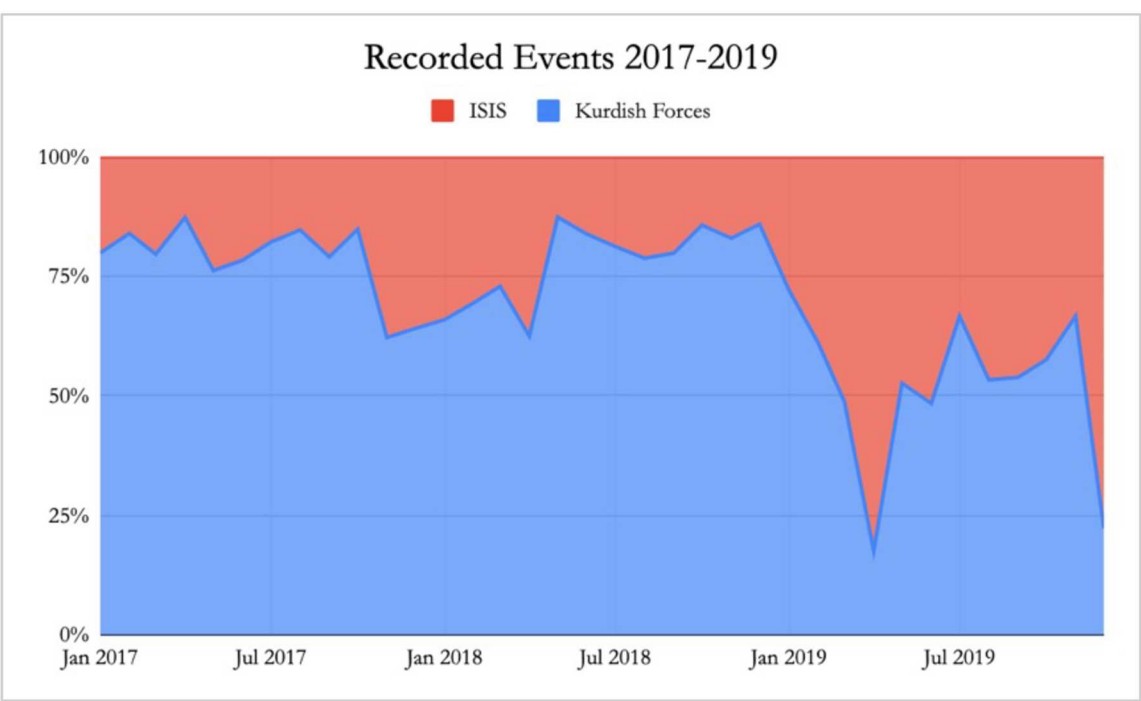

**Fig 6. Proportion of total events per actor.** Data source: [100]. Proportion of events recorded by ACLED in the ISIS-Kurdish conflict carried out by each actor in the conflict. As Kurdish forces gain control of territory, particularly towards 2019, we observe a larger proportion of events carried out by ISIS.

emphasise direct, bilateral engagements and immediate retaliations. Surprisingly, the model uncovers that a large majority of the behaviour consists of periods of one-sided violence, where retaliation, if present, is not immediate. Whilst we may expect one-sided violence due to offensives during the conflict, the low levels of observed retaliation are somewhat surprising.

The lack of immediate retaliation and the prevalence of one-sided violence underscore the complexity of strategy combinations in conflict. These combinations range from non-reactive strategies, where actions are independent of the opponent's moves, to reactive strategies that are contingent on the opponent's behaviour. However, the reliance on datasets like ACLED, which primarily record instances of conflict, limits the ability to fully dissect these strategy combinations within the Rojava context. The need for more comprehensive data, particularly encompassing aspects of cooperation and negotiation, is evident to enhance understanding of de-escalation and conflict resolution strategies. Significantly, territorial control emerges as the most critical factor influencing changes in conflict behaviour. This finding suggests that shifts in control of specific locations often precede surges of one-sided violence, challenging the traditional emphasis on temporal and geographical factors in conflict analysis. However, the variance in the intensity of violence required to effect territorial change highlights that locations cannot be viewed in isolation; the broader context is key to understanding these events.

The study also brings into focus the unique nature of the conflict between two non-state actors, diverging from the common focus on state versus non-state interactions in the literature. The interaction between an ethno-nationalist group and a religious fundamentalist group presents distinct strategic behaviours that may not be generalisable across different actor types. This raises intriguing questions about the consistency of strategic behaviour in conflicts involving similar groups and the influence of perceived legitimacy on the likelihood of conflict

outbreak and resolution, as suggested by Esteban and Ray [12], and Epstein [44]. The limitations presented by data availability and potential biases are acknowledged, tempering the generalisability of the findings. Despite cross-checking with other sources, data and reporting biases inherent in conflict situations [35, 65, 70] pose challenges to the accuracy and comprehensiveness of the analysis. Consequently, while the model offers valuable insights, its application in forecasting scenarios or generalisation to other conflicts is constrained.

The research presented here not only contributes to the current understanding of civil wars involving non-state actors but also opens several avenues for future research. One potential area is the exploration of similar dyadic interactions in different geopolitical contexts, which could reveal whether the strategic behaviours observed in the Kurdish-ISIS conflict are unique or if they exhibit universal patterns across non-state conflicts. Additionally, extending the model to include multi-party conflicts could offer insights into the complexities of larger scale civil wars, where alliances shift and multiple objectives intersect. Another promising direction is the integration of more granular, real-time data, especially encompassing aspects of negotiation and cooperation, to provide a more comprehensive picture of conflict dynamics. This could lead to the development of more sophisticated predictive models for conflict resolution and peacekeeping efforts. Furthermore, comparative studies of conflicts involving state versus non-state actors, or different types of non-state actors, could deepen our understanding of how ideological and organisational structures influence strategic decisions in conflicts. Finally, exploring the application of this modelling approach to post-conflict scenarios could yield valuable insights into the long-term effects of conflict on societal structures and the potential pathways to sustainable peace.

## Supporting information

**S1 Table. Game theory strategies from the literature [96].**
(PDF)

**S2 Table. Additional game theory strategies.**
(PDF)

**S3 Table. Strategy categorisation.** Resulting patterns of defection after playing strategies against all others and itself in an Iterated Prisoner's Dilemma game using the Axelrod library.
(PDF)

**S1 Appendix. Data extraction and pre-processing.**
(PDF)

**S2 Appendix. Implementation of Axelrod library.**
(PDF)

## Acknowledgments

The authors would like to thank Clare Lewis and Nils Metternich for their valuable suggestions and comments on this work.

## Author Contributions

**Conceptualization:** Olivia Macmillan-Scott, Akin Ünver, Mirco Musolesi.

**Data curation:** Olivia Macmillan-Scott.

**Formal analysis:** Olivia Macmillan-Scott.

**Methodology:** Olivia Macmillan-Scott, Akin Ünver, Mirco Musolesi.

**Supervision:** Akin Ünver, Mirco Musolesi.

**Visualization:** Olivia Macmillan-Scott.

**Writing – original draft:** Olivia Macmillan-Scott.

**Writing – review & editing:** Akin Ünver, Mirco Musolesi.

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
