## [Decision Letter · Decision Letter 0]

15 Oct 2023

PONE-D-23-18247Game-theoretic agent-based modelling of micro-level conflict: Evidence from the ISIS-Kurdish WarPLOS ONE

Dear Dr. Macmillan-Scott,

Thank you for submitting your manuscript to PLOS ONE. After careful consideration, we feel that it has merit but does not fully meet PLOS ONE’s publication criteria as it currently stands. Therefore, we invite you to submit a revised version of the manuscript that addresses the points raised during the review process.

Thank you for submitting your manuscript “Game-theoretic agent-based modelling of micro-level conflict: Evidence from the ISIS-Kurdish War” to PloS One. Your submission has undergone a thorough review process and I have received feedback from two reviewers.

One of the reviewers is quite positive about your work, appreciating the innovative application of game-theoretic and agent-based modelling in conflict analysis. They have indicated that your manuscript provides insightful and novel perspectives on analyzing conflict dynamics and have recommended acceptance of your manuscript.

On the other hand, another reviewer, while recognizing the potential in your ideas and approach, has raised several concerns that merit consideration. Their critique primarily revolves around clarity in the purpose, methodologies, and application of game-theoretical principles in your manuscript, as well as the specificities and relevance of the chosen case study. They also pointed out potential limitations in the data analysis and interpretation, and have recommended a more transparent and logically coherent presentation of the same.

While the feedback received is varied, both reviewers acknowledge the potential and relevance of your work, which aligns with the scope and interest of the journal. However, after reviewing your manuscript and considering the feedback provided by both reviewers, I concur with the concerns raised regarding the clarity and methodological explanation in your manuscript.

Therefore, the decision at this stage is “Minor Revision”. We encourage you to address the concerns and suggestions provided by the reviewers, particularly focusing on the (i) clarification of purpose, (ii) an explanation of your methodology, ensuring transparency and logical coherence in how the strategies from the Axelrod tournament and data from the ISIS-Kurdish conflict are applied and analyzed, (iii) elaborate on the choice of your case study, explaining its relevance and applicability for demonstrating your proposed analytical approach, and ensuring that its limitations and implications are transparently discussed.

We appreciate your submission to Plos One and believe that your research holds potential for valuable contributions to our readership. I look forward to receiving your revised manuscript and moving forward towards publication.

Sincerely,

José Manuel Galán

We look forward to receiving your revised manuscript.

Kind regards,

José Manuel Galán, Ph.D.

Academic Editor

PLOS ONE

“This work was partially supported by The Alan Turing Institute under the UK EPSRC grant EP/N510129/1. The authors would like to thank Nils Metternich for his valuable suggestions and comments on this paper.”

“This work was partially supported by The Alan Turing Institute

(https://www.turing.ac.uk/) under the UK EPSRC grant EP/N510129/1 (A.U. and M.M.).

The funders had no role in study design, data collection and analysis, decision to

publish, or preparation of the manuscript.”

Reviewers' comments:

Reviewer's Responses to Questions

**Comments to the Author**

1. Is the manuscript technically sound, and do the data support the conclusions?

Reviewer #1: Partly

Reviewer #2: Yes

2. Has the statistical analysis been performed appropriately and rigorously? 

Reviewer #1: I Don't Know

Reviewer #2: Yes

3. Have the authors made all data underlying the findings in their manuscript fully available?

Reviewer #1: Yes

Reviewer #2: Yes

4. Is the manuscript presented in an intelligible fashion and written in standard English?

Reviewer #1: Yes

Reviewer #2: Yes

5. Review Comments to the Author

Reviewer #1: I have been asked to evaluate the manuscript “Game-theoretic agent-based modelling of micro-level conflict: Evidence from the ISIS-Kurdish War”. The manuscript appears to be a second submission, since there is a revision memo attached to the end (I have not seen the original reviews, only the comments quoted in the responses). I am not entirely sure if the manuscript submitted is the revision version, since the manuscript ID does not have an R1 number, and some of comments echoed some reactions I had to reading the manuscript.

As far as I can see this manuscript uses event data for ISIS-Kurdish conflict over the period 2017-19, and seeks to compare them to strategies in an iterated prisoners dilemma. As far as I understand, Figure 1 compares responses for a pairwise list of a series of strategies (based on the Axelrod tournament?). We then get a delineation of a series of tables of the data, including “frequency of patterns” by location and recorded events, and the authors conclude that “looking at the proportion of attacks carried out by

each side is a better indicator of significant changes in the conflict than specific patterns of strategic interaction”.

I am very sympathetic to the ideas expressed at the outset - namely that game theory can inform empirical analyses of conflict and that computational models can benefit from engaging with empirical data – but I must confess that I found this manuscript difficult to understand. It is unclear to me exactly what the manuscript seeks to do, what it does, and what we could learn from this, echoing the previous comments by R2. Assuming that what I have seen is the revised version and not the original, I would say that the revisions have not succeeded in clarifying the purpose or analysis.

Some comments that may help clarify the contribution

The article needs a clearer purpose that “understanding the interaction between actors in conflict”. I.e., there needs to be something more than just a description of the data, for example predicting type of events/trajectories, and some comparison of a proposed approach against alternatives (e.g., predictions from a strategic model against autoregressive trends, for example).

The specific postulated contribution from game theory needs to be clarified. There are many analyses that seek to examine evidence for reciprocity in conflict levels (e.g., Goldstein and Pevehouse, Goldstein and Freeman), and could be seen as models to understand the effects of behaviors and possible interventions. The approach here seems to do something else – take a list of possible strategies in the Axelrod tournament, and then see if we can find evidence that the observed case resembles this. If if understand this correctly, the authors conclude that this cannot be done, since different strategies would lead to the same observed outcomes (i.e., pp.6-7). But the authors have not really clarified what it is they want to do and how this would be helpful.

I appreciate that iterated PDs could be seen as a model of continuing conflict, which could escalate up and down, in dyadic interactions under receiprocity with different event sequences. But the application here is rather confusing since the authors talk about events as C and D, but then say that ACLED only allows them to measure Ds, and the data are split up by location and seemed to be binnned in time. The analysis simply seems to count number of attacks, and assume that responses must be place specific and have no memory of past interactions. I am not sure exactly what is done here or what we are learning from this. The results seem to be largely a description of the observed data, without any clear added value from a strategic perspective.

I am also not sure why this is a particularly useful case to demonstrate a new approach to civil war interactions. It is a single dyadic interaction in a large civil war, unusual in that it is two non-state actors fighting rather than a recognized government (as is the case in typically very asymmetric civil wars). But presumably the interactions in such a setting will also be influenced by other actors in the conflict. If the aim is to examine a simple example, why not chose a conflict limited to two actors?

I am sorry that I could not give a more positive review of the current manuscript, but I hope these comments may help illustrate some of the reactions a potentially interested reader may have and the revisions called for to make the manuscript more accessible.

Reviewer #2: The article introduces an empirically grounded, agent-based model of civil conflict that is informed by game theory. Utilizing data from the conflict between Kurdish forces and the Islamic State in Rojava, Syria, the study aims to elucidate the interactions among conflict actors. The findings underscore that shifts in territorial control are more salient factors in the dynamics of the conflict than geographical or temporal variables. Moreover, the conflict interactions are predominantly characterized by episodes of unilateral violence, suggesting that the distribution of attacks holds greater relevance than specific behavioral patterns.

The authors have adequately addressed the concerns raised previously by peer reviewers. The research objective is clearly articulated, and the outcomes are insightful and meet the journal's criteria for publication.The proposed approach constitutes a advancement in the field of conflict analysis by offering a tool designed to deepen in the understanding of the dynamics of local strategies in wartime contexts.

6. PLOS authors have the option to publish the peer review history of their article (what does this mean?). If published, this will include your full peer review and any attached files.

Reviewer #1: No

Reviewer #2: No

---

## [Author Response · Author response to Decision Letter 0]

24 Nov 2023

Reviewer 1

1. The article needs a clearer purpose that “understanding the interaction between actors in conflict”. I.e., there needs to be something more than just a description of the data, for example predicting type of events/trajectories, and some comparison of a proposed approach against alternatives (e.g., predictions from a strategic model against autoregressive trends, for example).

We would like to thank the reviewer for their comment, as this made us revisit how we were presenting our research and placing it in the context of existing literature. We have made changes particularly to the Abstract, Introduction and Conclusion to improve clarity throughout, as mentioned in our response to the academic editor’s first point. We clarified the fact that our contribution goes beyond the mere description of the data. In fact, the game-theoretic approach allows us to get a better understanding of the dynamics of interaction within this conflict. We agree that these points were not sufficiently clear in the previous version of the manuscript. This is highlighted in the conclusion, and we have made this clearer in lines 603-608:

“Our model reveals that the proportion of attacks carried out by each side is a more indicative measure of significant changes in the conflict than previously recognized patterns of strategic interaction. This finding challenges conventional approaches in conflict analysis, which often emphasise direct, bilateral engagements and immediate retaliations. Surprisingly, the model uncovers that a large majority of the behaviour consists of periods of one-sided violence, where retaliation, if present, is not immediate.”

We do not compare our model to approaches based on regression and more statistical analyses - the reason for this is the fact that, as mentioned in the article, the majority of these approaches focus on studying the causes and onset of conflict. Instead, we are interested in the dynamics during an ongoing conflict.

2. The specific postulated contribution from game theory needs to be clarified. There are many analyses that seek to examine evidence for reciprocity in conflict levels (e.g., Goldstein and Pevehouse, Goldstein and Freeman), and could be seen as models to understand the effects of behaviors and possible interventions. The approach here seems to do something else – take a list of possible strategies in the Axelrod tournament, and then see if we can find evidence that the observed case resembles this. If if understand this correctly, the authors conclude that this cannot be done, since different strategies would lead to the same observed outcomes (i.e., pp.6-7). But the authors have not really clarified what it is they want to do and how this would be helpful.

As mentioned in our response to the academic editor’s third point, part of our contribution comes from the types of actors involved in the chosen case study. The literature mentioned by the reviewer (Goldstein and Pevehouse, Goldstein and Freeman, etc.) looks at international conflicts and aggregates actors on a country level. Instead, we are looking at the interaction between two non-state actors that are equally matched, which lends itself to the application of the Prisoner’s Dilemma.

To address the point raised by the reviewer with respect to the objective of the study, we have significantly revised the abstract, introduction and conclusion. For instance, in lines 9-14 we emphasise the aim of the research:

“By integrating game theory into the agent-based framework, we scrutinise the evolving nature of violent events in Rojava, examining whether these incidents follow temporally and spatially stable patterns or vary significantly over time and location. This approach not only fills a notable gap in conflict literature but also provides a sophisticated understanding of the strategic behaviours characterising the Kurdish-ISIS conflict, offering insights into broader conflict dynamics and resolution strategies.”

3. I appreciate that iterated PDs could be seen as a model of continuing conflict, which could escalate up and down, in dyadic interactions under receiprocity with different event sequences. But the application here is rather confusing since the authors talk about events as C and D, but then say that ACLED only allows them to measure Ds, and the data are split up by location and seemed to be binnned in time. The analysis simply seems to count number of attacks, and assume that responses must be place specific and have no memory of past interactions. I am not sure exactly what is done here or what we are learning from this. The results seem to be largely a description of the observed data, without any clear added value from a strategic perspective.

The reviewer is correct in that the ACLED data only allows us to study instances of defection, which we discuss in the Modelling Approach section. However, we would like to point out that the analysis does not simply count the number of attacks - we look at the patterns present in the sequences of conflict events, and match these patterns to those that emerge from Prisoner’s Dilemma strategies. While we do carry out the analysis in specific locations, looking at the patterns within each location rather than across all recorded events, we also include a discussion of general trends in the data. 

As mentioned above in our response to the reviewer’s first point, we do not provide only a description of the observed data. Our model reveals a lack of immediate retaliation in most instances, where a large majority of the behaviours observed are made up of one-sided violence. We also find that the proportion of attacks carried out by each side is a more indicative measure of significant changes in the conflict than previously recognized patterns of strategic interaction. 

4. I am also not sure why this is a particularly useful case to demonstrate a new approach to civil war interactions. It is a single dyadic interaction in a large civil war, unusual in that it is two non-state actors fighting rather than a recognized government (as is the case in typically very asymmetric civil wars). But presumably the interactions in such a setting will also be influenced by other actors in the conflict. If the aim is to examine a simple example, why not chose a conflict limited to two actors?

We have addressed this question in our response to the academic editor’s third point. As the reviewer mentions, this case is unusual in that it involves two non-state actors. We also agree with the observations of the reviewer. We are focusing on actors that are closely matched in terms of power. This is indeed in contrast with an interaction between a state actor and a non-state actor, which is generally more asymmetric. We also chose to focus on the particular region of Rojava due to the sustained conflict between the two actors we study, and the lack of changes in alliances - as we have seen, shifting alliances are typical in the context of civil conflict, and the absence of this makes the chosen case more appropriate to study dyadic interaction. Therefore, to address the reviewer’s final question, by focusing on the region of Rojava, we have indeed chosen a conflict limited to two actors. We hope that the rewording of parts of the introduction and conclusions will contribute to clarify these points.

Reviewer 2

We would like to thank Reviewer 2 for their time and for their positive feedback on the paper.

---

## [Editor Report · Decision Letter 1]

8 Jan 2024

Game-theoretic agent-based modelling of micro-level conflict: Evidence from the ISIS-Kurdish War

PONE-D-23-18247R1

Dear Dr. Macmillan-Scott,

We’re pleased to inform you that your manuscript has been judged scientifically suitable for publication and will be formally accepted for publication once it meets all outstanding technical requirements.

Kind regards,

José Manuel Galán, Ph.D.

Academic Editor

PLOS ONE

Additional Editor Comments (optional):

After the revisions made to the manuscript, I am pleased to accept the article for publication in PLOS ONE. The changes have significantly improved the clarity and rigor of the work, ensuring that it meets the high standards of our journal. This decision reflects our confidence in the quality and relevance of your research to our readership. Congratulations on this accomplishment, and we look forward to sharing your valuable insights with the scientific community.
---

## [Editor Report · Acceptance letter]

11 May 2024

PONE-D-23-18247R1 

PLOS ONE

Dear Dr. Macmillan-Scott, 

I'm pleased to inform you that your manuscript has been deemed suitable for publication in PLOS ONE. Congratulations! Your manuscript is now being handed over to our production team.

Kind regards, 

on behalf of

Dr. José Manuel Galán 

Academic Editor

PLOS ONE